# Finding and Characterising Exceptional Distributions with Neural Rule Extraction

## Abstract

Explaining the characteristics of patients with an unusual disease mortality can be an important tool to a clinician to understand and treat diseases. More generally, our goal is to find subsets of the data where the distribution of the target property— e.g. patient survivability— differs. The discovered subset must also defined by a human-interpretable rule over some given descriptive features. However, previous methods typically constrain the property of interest to be a scalar, which must also follow some standard distribution. Additionally, they require a prohibitive computational complexity for larger number of features, while, invariably, applying them on numerical features requires their a-priori discretisation.

To this end, we propose SYFLOW, a method which leverages the flexibility of normalising flows to learn any distribution that the property of interest may follow. With this, we then quantify the KL-divergence of this distribution in the discovered subset, thus yielding an objective that can be directly optimised all the way back to learnable feature weights. These, in turn, result in interpretable descriptions like "*Patients with heart disease and blood cholesterol above 243mg/dL*".

When applied on established real-world datasets, SYFLOW provides easily interpretable descriptions in a fraction of the times of state-of-the-art methods, and seamlessly extends onto multi-variate target properties, such as images. In evaluating on synthetic datasets, we also outperform the competition, when the target property does not follow a simple distribution. In general, SYFLOW enables to find notable trends in their data in a wide range of applications.

## 1 Introduction

Let us momentarily step into the shoes of a researcher that has just come across a novel dataset and tries to understand its inner workings. The, typically, numerous entries of the dataset necessitate automated tools that can shed light on its particularities. However, even though a multitude of machine learning methods exist to model the global distribution over all available entries, it is often important to be able to find only a subset of them: one that behave differently—a task that global methods have been shown to easily overlook (Kalofolias, 2023).

For instance, when we study patient medical data it becomes important to know which of the patients have an atypical disease progression: vulnerable minorities (Zeberg & Pääbo, 2020) could receive additional support from the state or more resilient ones could reveal relevant biomarkers that can help combat the disease (Goossens et al., 2015; Beigel et al., 2020). In sociology, local deviation of welfare metrics can pinpoint disadvantaged demographic groups (Boll & Lagemann, 2019; Ortiz & Cummins, 2011), while in materials science it is of core importance to find a combination of properties that single out a collection of desirable materials (Sutton et al., 2020). Especially, in such applications the number of features can easily exceed the thousands, while, for instance, when it comes to omics processing and the analysis of the respective data, the number of candidate properties easily reaches the order of tens of thousands or more. Yet, only a handful of mutations may be enough to single out a small set of drug-resistant species. Importantly, the very same tools can also aid any user of black box machine learning models, when we need to scrutinise the model for any subset of the data where it behaves differently—be it with a lower accuracy on a local, difficult subset of the data, or, before it is used on individuals, a set of them on which it could introduce demographic disparity (Zachary Izzo & Zou, 2023).

In the common denominator of all aforementioned applications lies a need to present the relevant discoveries to a human end-user. In other words, not only do we require that we discover subsets with exceptional behavior, but also, and above all, that they can clearly be interpreted by the respective audience. These two conditions, together, give rise to the task of subgroup discovery: the discovery of a simple description that corresponds to a subset in the data, in which some property of interest behaves locally exceptionally when contrasted to the entire dataset. Typically, such a description is adopted to be a simple conjunction of predicates, each of which is based on the available features of the entities under study. For instance, when the property of interest is such a description for medical patients could be of the kind: "*Patients with heart disease and blood cholesterol above* $243mg/dL$".

In its history of more than three decades (Klösgen, 1996), several approaches have been proposed for subgroup discovery (Atzmueller, 2015). However, they have arguably not kept up with the recent advances in machine learning, which involve not only a large number of features, but also the use of arbitrary distributions that must be directly learned from the data. Indeed, most previous methods assume that the property of interest follows a standard distribution, for instance a normal or a binomial, and is typically limited on only scalar properties. Other works use a proxy instead of learning the distribution of the property of interest, which renders their results less intuitive. Additionally, typical solutions only approach this task from the standpoint of combinatorial optimisation, which often takes a significant toll on the required run times, especially when it comes to the exact optimisation of the respective objectives. As a result of the combinatorial explosion of the search space as the number of predicates increases, before any of the previous methods can be applied on continuous features, it requires a quantisation of the features into a few simple predicates each. As we show, this quantisation can greatly reduce the quality of the results.

Hence, we focus in the usual case of continuous features, and break with the combinatorial treatment of this task, whatsoever. Instead, we derive a differentiable objective, which allows us to directly learn the distribution of the target property in the dataset by fitting a normalising flow (Rezende & Mohamed, 2015). We also use a soft rule to describe candidate subgroups, which depends on a learnable lower and an upper bound per feature. Thus, as we tune these bounds for each feature during optimisation, we consider all possible simple predicates per feature, out of which in previous methods only a few were available after the quantisation step. These two components work in tandem, to be able to detect possible deviations in the global distribution of our property of interest, while at the same time adjusting the soft rule into a ever crisper rule, that in the end results in an explicit, human-interpretable rule.

A key characteristic of our method is its scalability, when contrasted with exact methods. This fact is largely due to using a differentiable optimisation objective for this task, which allows the fast convergence to a good local optimum. In addition, as this enables us to use the mature stochastic optimisation machinery, we can benefit also from the existence of highly optimised hardware for such computations; for instance, our algorithm can readily be offloaded to multiple graphics processing unit cores, in parallel.
Overall, the main advantages of our method, SYFLOW, can be summarised in its ability

    (i) to differentiate between *arbitrary distributions* for the target property within the subgroup,
   (ii) to search over all predicates generated from each feature, avoiding their quantisation, and
  (iii) to scale readily to orders of magnitude larger datasets, even in terms of number of features.

## 2   PRELIMINARIES

We consider a dataset of $n$ pairs $(\mathbf{x}, y)$, where $\mathbf{x} \in \mathbb{R}^p$ is a real-valued **feature vector** and $y \in \mathcal{Y}$ a property of interest, the **target property**. Our goal is to find rules over the values of $\mathbf{x}$ that specify such a subset of the dataset, in which the distribution of the target property $y$ differs from that of the entire dataset. As a **rule** we consider a Boolean-valued function $\sigma : \mathbb{R}^p \rightarrow \{0, 1\}$, which specifies whether a pair belongs to the subset or not. The rules we consider take the form $\sigma : \mathbf{x} \mapsto \wedge_{i=1}^{p} \pi(x_i; \alpha_i, \beta_i)$, i.e., they are the conjunction of one predicate per feature. The **predicates**, themselves, are Boolean-valued functions $\pi : x \in \mathbb{R} \mapsto \mathbb{1}\{\alpha < x < \beta\}$, which are true when the value of the specific feature lies in the interval $(\alpha, \beta)$. Here, we used the indicator function $\mathbb{1}\{\cdot\}$, which yields 1 (true) whenever the condition $\cdot$ is true, and 0 (false), otherwise. We hence call a **subgroup** any subset of the dataset that can be formed by applying one of our rules over it. This implies that a subgroup is thus endowed with a human-interpretable description. For instance, in

a dataset of patients with the two features *age* and *weight*, the rule $\pi(x_1; 13, 19) \wedge \pi(x_2; 80, 100)$ defines the subgroup "*teenager patients with weight between 80 and 100*".

From a statistical perspective, we assume that $(\mathbf{x}, y)$ is a realisation of a pair of random variables $(\mathbf{X}, Y) \sim \mathbb{P}(\mathbf{X}, Y)$. We further introduce a binary random variable $S \in \{0, 1\}$ that denotes membership to a subgroup whenever $S = 1$. Hence, we seek for a rule that, among other criteria, maximises a measure of dissimilarity between the distributions $\mathbb{P}(Y)$ and $\mathbb{P}(Y|S = 1)$. Here, for random variables we use capital symbols, for their densities a small $p$, and for their laws a capital $P$.

## 3 A Differentiable Objective for Exceptional Subgroups

We here formalise our goal into a differentiable objective that can be optimised within the framework of stochastic gradient descent. At the core of this objective lie two key components: 1) a measure of dissimilarity between the two distributions $p_{Y|S=1}$ and $p_Y$, which depends on 2) a differentiable estimation of the subgroup membership probability $s(\mathbf{x}) \approx \mathbb{P}(S = 1|\mathbf{X} = x)$. In the following we present these parts and we finally compose them into our complete objective.

### 3.1 Differentiable Rule Induction

For the sake of differentiability, instead of modelling the subgroup membership $\sigma$, we instead model its probability. That is, from the statistical inference perspective, we assume that $S|\mathbf{X} = \mathbf{x} \sim \text{Bernoulli}\left(\sigma(\mathbf{x}; \alpha, \beta)\right)$, for $\alpha, \beta \in \mathbb{R}^p$; we then seek $\hat{\alpha}, \hat{\beta} \in \mathbb{R}^p$ so that $s(\mathbf{x}; \hat{\alpha}, \hat{\beta}) \approx \sigma(\mathbf{x}; \alpha, \beta)$.

To compose a model for $s$ we follow the derivation of the crisp membership function $\sigma$, replacing each part with an appropriate continuous analogue; we further do so in a way that allows the derivation of a clear description from the parameters of the function $s$ after its learning. We hence need to provide such analogues for 1) the per-feature predicates $\pi(x; \alpha, \beta)$ and 2) their conjunction. Due to the logical reasoning in our task, we use concepts from differentiable rule learning (see Sec. 4.1).

Hence, as a differentiable analogue to the per-feature predicate we adopt the *soft-binning* function introduced by Yang et al. (2018), which in its original form uses an one-hot bin encoding. For our needs we adapt it to use a lower and upper bound $\alpha, \beta \in \mathbb{R}$, as well as a temperature parameter $t > 0$ that controls the binning steepness. Thus, to each feature we associate the **soft-binning** predicate

$$\pi(x_i; \alpha_i, \beta_i, t) = \frac{\exp{(2x_i - \alpha_i)}^{1/t}}{\exp{(x_i)}^{1/t} + \exp{(2x_i - \alpha_i)}^{1/t} + \exp{(3x_i - \alpha_i - \beta_i)}^{1/t}} \; . \tag{1}$$

Importantly, in the limit as $t \to 0$ each soft-binning predicate converges to a crisp thresholding rule.

**Theorem 1** *Given its lower and upper bounds $\alpha_i, \beta_i \in \mathbb{R}$, the soft-binning predicate of Eq. (1) applied on $x \in R$ converges to the crisp rule function that decides whether $x \in (\alpha, \beta)$,*

$$\lim_{t \to 0} \hat{\pi}(x_i; \alpha_i, \beta_i, t) = \begin{cases} 1 & \text{if } \alpha_i < x_i < \beta_i \\ 0.5 & \text{if } x_i = \alpha_i \vee x_i = \beta_i \\ 0 & \text{else} \end{cases} \; .$$

We provide the full proof for the general case with $M$ bins in the Appendix B.

The soft-binning predicate provides a differentiable, adaptable binning function from which we construct the fuzzy predicate $\hat{\pi}$. The rule function $\sigma(x)$ is a logical conjunction of all fuzzy predicates. To avoid the problem of *vanishing gradients* when utilizing a multiplication to model the logical conjunction, we instead propose to utilize the harmonic mean $\mathcal{M}(x) = \frac{p}{\sum_{i=1}^{p} \hat{\pi}(x_i; \alpha_i, \beta_i, t)^{-1}}$ to model the logical conjunction. The harmonic mean behaves as desired for strictly binary predicates, i.e. $\exists \hat{\pi}(x_i; \alpha_i, \beta_i, t) = 0 \Rightarrow \mathcal{M}(x) = 0$ and $\forall \hat{\pi}(x_i; \alpha_i, \beta_i, t) = 1 \Rightarrow \mathcal{M}(x) = 1$. Still, for high dimensional feature spaces $\mathcal{X}$, the harmonic mean may struggle when given many fuzzy predicates $\hat{\pi}(x_i; \alpha_i, \beta_i, t)$. Hence, we use the *weighted harmonic mean* to model the logical conjunction, and define the fuzzy logical-conjunction as

$$s(x; \alpha, \beta, a, t) = \frac{\sum_{i=1}^{p} a_i}{\sum_{i=1}^{p} a_i \hat{\pi}(x_i; \alpha_i, \beta_i, t)^{-1}} \; .$$

The weights $a \in \mathbb{R}^p$, which are constrained to be positive through a ReLU function, allow the conjunction layer to disable unnecessary predicates. The weights do not affect behavior for strictly binary predicates, wherever $a_i > 0$. In practice, we observe that the proposed conjunction $s$ finds good rules for both large logical conjunctions as well as large dimensionality $p$ of the data.

## 3.2 DENSITY ESTIMATION WITH NORMALISING FLOWS

Besides a differential rule function, our method requires an accurate density estimation of the target variable in the subgroup and in general. For this, we adopt *Normalising Flows*, an increasingly popular class of density estimators (Papamakarios et al., 2021). The fundamental idea behind Normalising Flows is to start with a distribution with a known density function, such as a Gaussian distribution with $p_\mathcal{N}$, and fit an invertible function $f$ to transform it onto the target density function.

The normalising flow architecture of our choice are Neural Spline Flows (Durkan et al., 2019), which use expressive yet invertible piece-wise, polynomial spline functions, although SYFLOW allows to seamlessly use any other normalising flow architecture, depending on the input domain. In general, the idea is to train the function $f$ so that $p_\mathcal{Y} \approx f(p_\mathcal{N})$. Given a sample $y$, we can compute the likelihood of that point under the current function $f$ as $p_{f(\mathcal{N})}(y) = p_\mathcal{N}(f^{-1}(y))|\det\left(\frac{\delta f^{-1}(y)}{\delta y}\right)|$. Thus, given a sample of $\mathbb{P}(Y)$, we can maximise the likelihood of $p_{f(\mathcal{N})}(y)$ and hence fit $p_{f(\mathcal{N})} \approx p_\mathcal{Y}$.

## 3.3 MEASURING DISSIMILARITY BETWEEN TARGET DISTRIBUTIONS

Next, we provide a measure of dissimilarity between the target distribution within the subgroup $p_{Y|S=1}$ and the global target distribution $p_Y$, both of which we will model using normalising flows later on. As a measure of their dissimilarity we adopt the Kullback-Leibler (KL) divergence

$$D_{\text{KL}}\left(P_{Y|S=1}\|P_Y\right) = \int_{y \in \mathcal{Y}} p_{Y|S=1}(y) \log\left(\frac{p_{Y|S=1}(y)}{p_Y(y)}\right) dy \, . \tag{2}$$

In Eq. (2), however, the dependency on our subgroup model $s$ does not explicitly appear, which is needed for its optimisation. To amend this, we rewrite the first occurrence of $p_{Y|S=1}$ in Eq. (2) as

$$p_{Y|S=1}(y) = \int_{\mathbf{x} \in \mathbb{R}^p} p_{Y|S=1,\mathbf{x}}(y,\mathbf{x}) p_{\mathbf{x}|S=1}(\mathbf{x}) d\mathbf{x} = \int_{\mathbf{x} \in \mathbb{R}^p} p_{Y|S=1,\mathbf{x}}(y,\mathbf{x}) \frac{p_{S=1|\mathbf{x}}(\mathbf{x}) p_{\mathbf{x}}(\mathbf{x})}{\mathbb{P}(S=1)} dx \, , \tag{3}$$

where we first used the rules of marginal probability and then that of Bayes. We now first approximate Eq. (3), and then show how we can estimate the KL divergence of Eq. (2) for its optimisation.

To this end, we first note that the latent subgroup indicator random variable $S$ takes only two discrete values, depending on whether $\mathbf{x}$ belongs to the subgroup or not; we use this to partition the domain of integration $\mathbb{R}^p$ into $\mathbb{R}^p_\in := \{\mathbf{x} \in \mathbb{R}^p | \sigma(\mathbf{x}) = 1\}$ and $\mathbb{R}^p_\notin := \{\mathbf{x} \in \mathbb{R}^p | \sigma(\mathbf{x}) = 0\}$. We next introduce three reasonable assumptions, under which we can approximate the density $p_{Y|S=1}$ of Eq. (3).

First, we assume that both $p_\mathbf{x}$ and $p_{Y|S=1,\mathbf{x}}$ are upper bounded by the finite constants $C_\mathbf{X} > 0$ and $C_Y > 0$, respectively. Secondly, we assume that in a subset $\mathbb{R}^p_{\notin\subset} \subset \mathbb{R}^p_\notin$ the membership probability is negligible and, lastly, that $\mathbb{R}^p_{\notin\subset}$ covers almost all of the non-membership domain $\mathbb{R}^p_\notin$. Formally,

$$p_\mathbf{x}(\mathbf{x}) \leq C_\mathbf{X}, \quad (4) \qquad \int_{\mathbf{x} \in \mathbb{R}^p_{\notin\subset}} p_{S=1|\mathbf{x}}(\mathbf{x}) d\mathbf{x} \leq \epsilon_1 \, , \quad (6) \qquad \int_{\mathbf{x} \in \mathbb{R}^p_\notin \setminus \mathbb{R}^p_{\notin\subset}} p_\mathbf{x}(\mathbf{x}) d\mathbf{x} < \epsilon_2 \, . \tag{7}$$
$$p_{Y|S=1,\mathbf{x}} \leq C_Y, \quad (5)$$

**Theorem 2** *Under the assumptions of Eq.* (4), (5), (6), *and* (7), *it is*

$$p_{Y|S=1}(y) - \int_{\mathbf{x} \in \mathbb{R}^p_\in} p_{Y|S=1,\mathbf{x}}(y,\mathbf{x}) \leq \frac{C_Y(\epsilon_2 + C_\mathbf{X}\epsilon_1)}{\mathbb{P}(S=1)} d\mathbf{x} \, .$$

*Further, during learning, this bound becomes tighter until it asymptotically vanishes, assuming a decreasing annealing schedule for the temperature parameter.*

We postpone the proof to Appendix A.

Using the same assumptions, we further approximate for all $\mathbf{x} \in \mathbb{R}^p_\in$ the target property conditional

$$p_{Y|\mathbf{x}}(y, \mathbf{x}) = p_{Y|S=1, \mathbf{x}}(y, \mathbf{x}) p_{S=1|\mathbf{x}}(\mathbf{x}) + p_{Y|S=0, \mathbf{x}}(y, \mathbf{x}) p_{S=0|\mathbf{x}}(\mathbf{x}) \approx p_{Y|S=1, \mathbf{x}}(y, \mathbf{x}), \quad \mathbf{x} \in \mathbb{R}^p_\in \ ,$$

which allows us to approximate the subgroup-conditional target distribution as

$$p_{Y|S=1}(y) = \int_{\mathbf{x} \in \mathbb{R}^p_\in} p_{Y|S=1, \mathbf{x}}(y, \mathbf{x}) \frac{p_{S=1|\mathbf{x}}(\mathbf{x}) p_{\mathbf{x}}(\mathbf{x})}{\mathbb{P}(S=1)} dx \approx \int_{\mathbf{x} \in \mathbb{R}^p_\in} p_{Y, \mathbf{x}}(y, \mathbf{x}) \frac{p_{S=1|\mathbf{x}}(\mathbf{x})}{\mathbb{P}(S=1)} dx \ . \quad (8)$$

Finally, we replace Eq. (8) into Eq. (2) to obtain our final approximation

$$D_{\mathrm{KL}}(P_{Y|S=1} \| P_{Y|S=1}) = \int_{y \in \mathcal{Y}} \int_{\mathbf{x} \in \mathbb{R}^p_\in} p_{Y, \mathbf{x}}(y, \mathbf{x}) \frac{p_{S=1|\mathbf{x}}(\mathbf{x})}{\mathbb{P}(S=1)} dx \log\left( \frac{p_{Y|S=1}(y)}{p_Y(y)} \right) dy \ .$$

From this point onward we can use the standard Monte Carlo estimation of this integral, which gives

$$D_{\mathrm{KL}}(P_{Y|S=1} \| P_Y) \approx \frac{1}{\bar{s}} \sum_{k=1}^{n} s(\mathbf{x}^{(k)}) \log\left( \frac{p_{Y|S=1}(y^{(k)})}{p_Y(y^{(k)})} \right) \ ,$$

where $p_Y$ and $p_{Y|S=1}$ stand for the models trained from the normalising flows, $s$ is our subgroup membership model (see Sec. 3.1) and $\bar{s}$ is estimated as $\frac{1}{n} \sum_{i=1}^{n} s(\mathbf{x}^{(i)})$.

### 3.4 RULE GENERALITY

Lastly, we deal with some fine tuning of our main objective. We observe that only maximising our measure of statistical dissimilarity does not necessarily lead to general rules. In fact, we could easily craft a small subgroup consisting of the most deviating sample on its own, defined by a rule with a very narrow scope and relatively low value. In light of this, we employ a common technique in subgroup discovery (Boley et al., 2017) in order to steer the results of the optimisation toward larger subgroups: we multiply the statistic of dissimilarity with the size of the subgroup $\bar{s}^\gamma$. The power $\gamma$ tunes the trade-off in the importance between the two factors.

Hence, summarising all the above, we obtain as our final objective our variant of the **size-corrected KL** (van Leeuwen & Knobbe, 2011)

$$D_{\mathrm{WKL}}(P_{Y|S=1} \| P_Y) = \bar{s}^\gamma \hat{D}_{\mathrm{KL}}(P_{Y|S=1} \| P_Y) \ . \quad (9)$$

### 3.5 FULL MODEL

In the previous sections, we have detailed our rule learning architecture with differentiable thresholding and aggregation (Sec. 3.1), how to use Neural Spline Flows to obtain non-parametric density estimates (Sec. 3.2), and finally derived Objective (9), a size aware Kullback-Leibler Divergence that allows us to optimize our rule function $s(\mathbf{x})$ with gradient descent. From these components, we build our new architecture for **S**ubgroup discovery with neuro-s**Y**mbolic guided normalizing **FLOW**s, or short SYFLOW. Given a dataset $\{(\mathbf{x}^{(k)}, y^{(k)})\} \sim \mathbb{P}(\mathbf{X}, Y)_{k=1}^{N}$, SYFLOW undergoes the following three steps for each sample $(\mathbf{x}^{(k)}, y^{(k)})$:

1. *Feature Thresholding*: Initially, all continuous features $\mathbf{x}_i^{(k)}$ are thresholded using the soft-binning function described in Theorem 1 with learned parameters $\alpha_i$ and $\beta_i$. This operation yields a fuzzy predicate vector $\hat{\pi}(\mathbf{x}^{(k)}; \alpha, \beta, t) \in [0, 1]^p$.

2. *Subgroup Rule*: Subsequently, our rule learner employs weights $a_i$ to combine the individual predicates $\pi_i(\mathbf{x}_i^{(k)})$ into a rule $s(\mathbf{x}; \alpha, \beta, a, t)$ using the harmonic mean. This rule represents the probability of $\mathbf{x}^{(k)}$ $p_{S=1|\mathbf{x}}(\mathbf{x}^{(k)})$ and is used to compute the KL-Divergence.

3. *Distribution Difference*: Lastly, we estimate the likelihood of $p_{\mathcal{Y}}(y^{(k)})$ and $p_{\mathcal{Y}|S=1}(y^{(k)})$ with two separately fitted normalising flow models. Then, according to Objective (9), we can estimate the KL-Divergence between the current subgroup and the general distribution.

By repeating the aforementioned steps over all samples $(\mathbf{x}^{(k)}, y^{(k)})$ and summing up the results, Objective (9) gives us a differentiable estimate of the KL-Divergence in regards to the subgroup

rule $s(\mathbf{x})$. We optimize $s(\mathbf{x})$ using standard gradient descent techniques with the Adam optimizer (Kingma & Ba, 2015). After the subgroup rule has been updated, we again update the normalising flow of the subgroups density as described in Sec. 3.2, and repeat this process for a user-specified amount of epochs. During the training, we gradually decrease the temperature $t$ by a pre-determined schedule to obtain increasingly crisp subgroup assignments. Finally, at the last epoch, the discovered subgroup is then the output of the subgroup rule $s(\mathbf{x})$. We provide a diagram overviewing and the pseudo-code for SYFLOW in the Appendix E.

## 4 RELATED WORK

For subgroup discovery (Klösgen, 1996) many approaches have been proposed over the years. These can largely be classified along two properties: their optimisation scheme and their subgroup quality measure. When it comes to the optimisation, the majority of methods follow a combinatorial treatment (Helal, 2016) and either use exact search (Atzmueller & Puppe, 2006) or are content with a locally optimal subgroup (Duivesteijn et al., 2016). However, despite any benefits of exact methods, since this task is NP-hard, these methods rarely scale beyond a few hundred features (Atzmueller & Puppe, 2006); this also highly depends on the input domain, e.g., whether the features are discrete or continuous. On the other hand, inexact methods, in which SYFLOW also belongs, can be applied on practical real-world settings with little discount in quality (Mandros et al., 2017). Here, other methods have used beam search and other heuristic combinatorial search variants (Duivesteijn et al., 2016), which still require a prior quantisation method for any real-valued features. In contrast, SYFLOW directly learning the per-feature bounds that form the predicates, obviating the need for quantisation, which leads to substantially improved quality (see Sec. 5.1).

When it comes to the quality measure, we do not treat methods that use subjective measures (Atzmueller, 2015), as they lack statistical soundedness and objective interpreetation. Indeed, these methods employ heuristics to define "subgroup interestingness" using arbitrary metrics of prior knowledge, like surprisal (Freitas, 1998). We hence focus on objective measures, which are based on statistical tests (Grosskreutz & Rüping, 2009) or measures of distribution dissimilarity (Song et al., 2016), on which our method also belongs. Out of these methods, those that perform exact search are limited on standard distributions such as normal (Friedman & Fisher, 1999; Lavrač et al., 2004), binomial or $\chi^2$ (Grosskreutz & Rüping, 2009), and as we show outperform ours in the quality of the results, when they can actually terminate. Invariantly, however, these methods are impractically slow, limit themselves to isolated distributions, and additionally require a quantisation step for continuous features. Out of those, we compare against both state-of-the-art methods on normal data.

Finally, inexact methods have been proposed toward arbitrary distributions, for which they introduce a proxy for dissimilarity. One line of work performs exceptional model mining (Duivesteijn et al., 2016): trains a model in the entire dataset and one within the subgroup; then it assesses dissimilarity by looking at the model weights. This approach lacks statistical interpretation and is prone to weaknesses of the simple models used. Proença et al. (2022) uses the minimum description length principle (MDL), which models the data distribution using a particular prior, then uses it as a proxy for KL-divergence. Although using this prior can be ill adapted to certain distributions, this latter work still remains the most related method of its kind to ours, and we compare against it. To the best of our knowledge, there have been no prior works that directly approximate the KL-divergence.

### 4.1 DIFFERENTIABLE RULE INDUCTION

Large part of our novelty lies in the use of a differentiable approach for our task. From this perspective, our work is related to differentiable rule induction methods: these also use a differentiable objective (Riegel et al., 2020), from which they finally extract a crisp rule (Wang et al., 2020; 2021).

Most related to our approach are works that model the logical operation of conjunctions and disjunctions with continuous analogues (Riegel et al., 2020; Fischer & Vreeken, 2021). There, for instance, the logical conjunction has been replaced with various $t$-norms: the product one, $x \wedge y := xy$ for the task of knowledge base reasoning (Yang et al., 2017) or that of Lukasiewicz, $x \wedge y := \max(0, x + y - 1)$ for semantic image interpretation (Donadello et al., 2017). In our work we propose a parameterised soft-multiplication that aims at alleviating the vanishing gradient in the decision boundary, while allowing for a tunable steepness as training progresses.

Importantly, other works use a neural architecture to learn the slabs of decision trees from data, which are then combined into a classification tree (Yang et al., 2018; Shi et al., 2022), that is end-to-end trained using gradient descent. In our work we adopt the learnable soft-binning technique of the former mutatis mutandis. All in all, however, our method is unique as it does not aim to learn simple rules to globally classify or regress a target property, but rather a rule that describes a subgroup with deviating distribution. This, to the best of our knowledge, is a novel contribution.

# 5 EXPERIMENTS

We evaluate SYFLOW against four state-of-the-art methods on synthetic and real-world data. We compare against Primp (PRIMP, Friedman & Fisher, 1999), subgroup discovery using mean shift (SD-$\mu$, Lemmerich & Becker, 2018), subgroup discovery using KL-divergence (SD-KL, Lemmerich & Becker, 2018) and Robust Subgroup Discovery (RSD, Proença et al., 2022).

## 5.1 SYNTHETIC DATA

To evaluate our methods on datasets with known ground truth we generate synthetic data. We conduct three experiments. First, we study the efficacy of the differentiable feature thresholding. In the second experiment we analyse the robustness towards various target distributions, and then we investigate the scalability of our methods regarding the number of features that the planted predicate spans, as well as the number of samples that it affects. As an evaluation measure we use the F1-score in terms of sample overlap between the discovered subgroup and the planted one as ground truth.

In all three experiments we start with an empty data matrix with 20000 rows and $p$ features. We sample uniformly from an $p$ dimensional cube with side length 1, i.e., $\mathbf{X} \sim \mathcal{U}(0, 1)^p$. The target $Y$ is initially sampled from $\mathcal{U}(0, 1)$. For the subgroup predicate we first sample $f_p$ features and then sample per feature an interval, such that the hypercube described by the predicate has volume 0.2.

**Feature Thresholding** Here, we fix the number of features at $p = 100$ and the length of the subgroup predicate at $f_p = 4$. In this experiment we only compare against SD-$\mu$, which is the best competing method for such data, and we vary the number of its cutpoints $cp \in \{2, 5, 10, 20, 30, 40, 50\}$.

As we can see in Figure 1b, as the number of cutpoints increase, the F1-score of SD-$\mu$ rapidly improves. However, only after $cp = 40$ does it achieve a slightly higher F1-score than SYFLOW. At the same time, as the number of cutpoints increases, the needed runtime increases rapidly (cf. Figure 1c). In particular, the runtime of SD-$\mu$ is $\sim 50$ times higher than that of SYFLOW.

**Complex Target Distributions** Similar to the first experiment, we use 100 features $p$ and set the subgroup predicate at a length of $f_p = 4$. We run experiments with five different target subgroup distributions for $\mathbb{P}(Y|S = 1)$, namely a Gaussian $\mathcal{N}(1, 0.5)$, a bi-modal mixture of Gaussians $0.5 \cdot \mathcal{N}(-1.5, 0.5) + 0.5 \cdot \mathcal{N}(1.5, 0.5)$, a beta distribution $\mathcal{B}(0.2, 0.2)$, which we scaled by 1.2, an exponential distribution $Exp(0.5)$ and a uniform one $\mathcal{U}(0.5, 1.5)$.

We present our results in Figure 1a; Here, we see that for distributions that are well characterized by their first moment, i.e., the uniform and normal distribution, SD-$\mu$ achieves a slightly higher F1-score than SYFLOW. In contrast, for more complex distributions, such as the bi-modal mixture of Gaussians or the exponential one, all methods except SYFLOW fail to recover subgroups. Moreover, the performance of SYFLOW is stable across all target distributions.

**Scalability** In the final experiment we vary both the number of features $f \in \{10, 50, 100, 250, 500, 750, 1000\}$, and the number of features used by the subgroup predicate $f_p \in \{4, 10\}$. The distribution in the subgroup now always follows a uniform distribution $\mathbb{P}(Y|S = 1) = \mathcal{U}(0.5, 1.5)$. The experiments are terminated after 24 hours. We present our results in Figure 2.

We observe that SYFLOW is significantly faster than all of its competitors, e.g., SD-$\mu$ takes 50 times longer to finish, while RSD could not finish already for more than 100 features. As expected, SD-$\mu$ achieves the highest F1-score, since the mean is a sufficient statistic to assess the difference between two uniform distributions. However, we significantly outperform all other competitors.

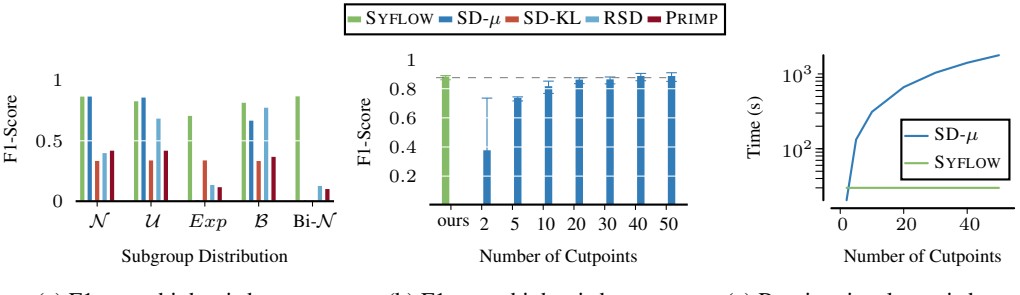

(a) F1-score higher is better    (b) F1-score higher is better.    (c) Runtime in s lower is better.

Figure 1: *Subgroup Predictive Capacity.* Method comparison in terms of F1-score against planted subgroup. Across different distributions: **(a)** SYFLOW outperforms the competition when these have higher order moments. Effect of number of cutpoints: as we allow more cutpoints on SD-$\mu$, it outperforms SYFLOW **(b)** after $40$; however, it already becomes significantly slower after just $5$ cutpoints, while to achieve similar performance to SYFLOW it requires $50$ times longer **(c)**.

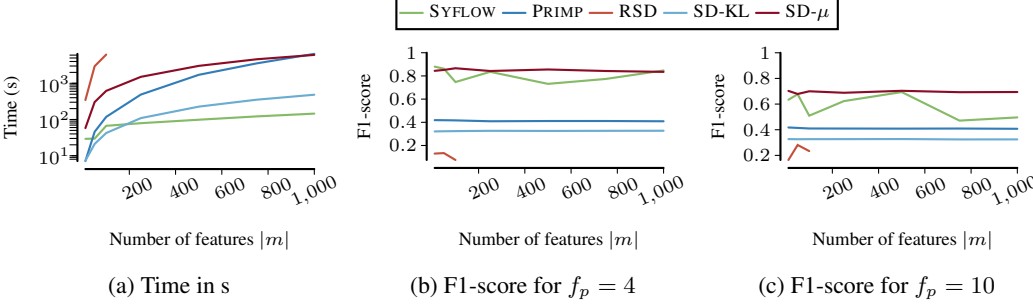

(a) Time in s    (b) F1-score for $f_p = 4$    (c) F1-score for $f_p = 10$

Figure 2: *Scalability.* In **(a)**, we plot the runtime as the number of features $m$ increases. SYFLOW is significantly faster than its competitors. In **(b)** and **(c)**, we plot the F1-score for $f_p = 4$ and $f_p = 10$.

## 5.2 REAL WORLD DATA

Next, we evaluate SYFLOW on 6 datasets from the UCI-Machine Learning Repository [1], all datasets of which are typically used for regression and data analysis tasks (Heskes et al., 2020; Wang et al., 2021; Heskes et al., 2020; Cortez et al., 2009). We study the same competitors as before. Since on real life datasets we have no access to ground truth subgroups, we report the *size-corrected* KL-divergence (SC-KL) of Eq. (9) between the found subgroup distribution and the overall population distribution, normalised by the size of the subgroup. In other words, we measure how dissimilar the distribution of the subgroup is, adjusted by its size. The results can be seen in Table 3a.

We can see that across all datasets, SYFLOW reliably finds subgroups whose target property distributions diverge from the overall one. That is, SYFLOW is either superior to or on par with its competitors in this metric. For instance, in the *Wages* dataset, SYFLOW significantly outperforms all other methods: as we see in Fig. 3b, the wage distribution resembles an exponential one, which justifies our superiority to other methods, in accordance with our earlier discoveries in Sec. 5.1.

## 5.3 CHARACTERIZING CLASSIFICATION ERRORS

Lastly, we perform a qualitative evaluation of SYFLOW by generating insight about the predictive accuracy of a classifier. We base our analysis on a Covid-19 ICU mortality dataset (Lambert et al., 2022) and train a random forest model for binary classification. On a separate test set, we take the predicted probabilities and compute the element-wise cross entropy to the ground truth labels as target variable $Y$. Our goal is to reveal demographic groups that our model is (not) able to handle especially well. The remaining biomarkers, which includes age, sex and pre-existing conditions, are used as the descriptive features $\mathbf{X}$.

---
[1]https://archive.ics.uci.edu/

| Dataset | SYFLOW | PRIMP | RSD | SD-KL | SD-$\mu$ |
|---|---|---|---|---|---|
| wine | **0.03** | 0.0 | 0.02 | 0.01 | 0.02 |
| insurance | **0.21** | 0.0 | 0.09 | 0.16 | 0.14 |
| wages | **0.12** | 0.0 | 0.02 | 0.03 | 0.03 |
| mpg | 0.24 | 0.07 | 0.22 | **0.32** | 0.26 |
| life | 0.22 | 0.09 | 0.06 | **0.23** | 0.22 |
| bike | **0.18** | 0.05 | **0.18** | 0.06 | 0.15 |
| Avg. Rank | **1.5** | 4.8 | 3 | 2.5 | 2.2 |

(a) Qualitative results for real world.

(b) Distributions of discovered subgroup.

Figure 3: *Real world*. In the Table we show, the results on various real-life datasets taken from the UCI-Machine Learning Repository. We report *size corrected* KL-divergence of the found subgroup and overall population. SYFLOW reliably finds subgroups that diverge from overall distribution.

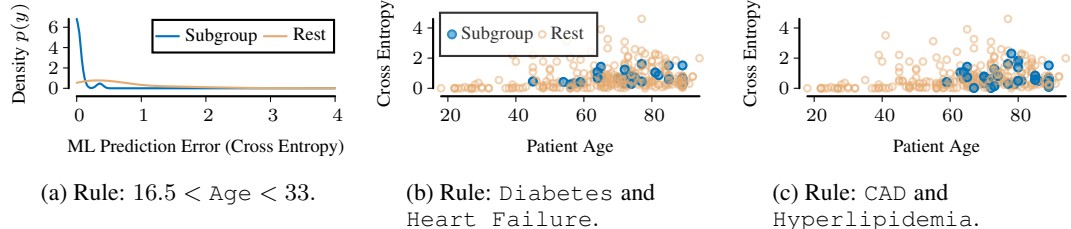

(a) Rule: $16.5 <$ Age $< 33$.

(b) Rule: Diabetes and Heart Failure.

(c) Rule: CAD and Hyperlipidemia.

Figure 4: *Covid-19 mortality prediction*. The classifier predicts correctly for young surviving patients (left). Accounting for age, the model classifies patients with both diabetes and heart failure more accurately (middle), and when they suffer from diabetes and coronary artery disease (right).

First, SYFLOW finds the subgroup characterized by the rule $16.5 <$ Age $< 33$. In fact, in this subgroup, the model predicts for younger people both confidently and correctly that they survive. This is reflected in the distribution of $p_{Y|S=1}$ with most of the probability mass in the low error region, displayed in Fig. 4a. This trend however is hardly surprising, but thanks to SYFLOW's flexibility, we instead opt to integrate age as a second variable into our analysis and re-run SYFLOW on the joint distribution of age and cross entropy.

Now, we find two main subgroups: "*patients with both diabetes as well as a history of congestive heart failure*" (Fig. 4b), and "*patients with high blood fat (hyperlipidemia) and coronary artery disease*" (Fig. 4c). In both cases, the model makes fewer errors, as in both subgroups significant risk factors for Covid-19 are known.

In principle, SYFLOW is applicable to any target domain with arbitrary dimensionality. In Appendix D we showcase a further example of SYFLOW's scalability by using images as target quantities $y$, where we discover subgroups that correspond to different classes of objects. Overall, the empirical evaluation demonstrates that SYFLOW characterizes subgroups with exceptional distributions in the widest range of settings, in terms of both speed and accuracy.

## 6 CONCLUSION

We proposed SYFLOW, an approach to discover locally optimal subgroups from a dataset, in which the distribution of the target property has a deviating distribution from that of the entire dataset. Our learned model consists of a component that approximates the membership probability of a sample in the subgroup and two normalising flows that gives the distribution of the target property in the dataset and subgroup, respectively. Importantly, the architecture of the membership probability model is chosen so as to allow the extraction of crisp, interpretable rules that define the subgroup. In experiments we show that our method is superior to the state of the art in at least scalability or predictive capacity for any of a large variety of target property distributions and number of features.

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

## A    PROOF OF THEOREM 3

**Theorem 3** *Under the assumptions of Eq.* (4)*,* (5)*,* (6)*, and* (7)*, it is*

$$p_{Y|S=1}(y) - \int_{\mathbf{x}\in\mathbb{R}^p_{\in}} p_{Y|S=1,\mathbf{x}}(y,\mathbf{x}) \leq \frac{C_Y(\epsilon_2 + C_{\mathbf{X}}\epsilon_1)}{\mathbb{P}(S=1)} d\mathbf{x} \,.$$

*Further, during learning, this bound becomes tighter until it asymptotically vanishes, assuming a decreasing annealing schedule for the temperature parameter.*

**Proof:**    *We first recall that, under our model,* $p_{S=1|\mathbf{x}}(\mathbf{x}) = \sigma(\mathbf{x}; \alpha, \beta)$ *for some* $\alpha, \beta \in \mathbb{R}^n$*, and is therefore a smooth function of* $\mathbf{x}$*. Intuitively, there are two regions of interest within* $\mathbb{R}^p_{\notin}$*: one within which it transitions from the value of almost 1 to that of almost 0, which is the region* $\mathbb{R}^p_{\notin} \setminus \mathbb{R}^p_{\notin \subset}$*, and a saturation region, where* $p_{S=1|\mathbf{x}} \to 0$ *super-exponentially, which is the region* $\mathbb{R}^p_{\notin \subset}$*. The particular thresholds that define these regions are not important, and any reasonable scheme leads to vanishing bounds* $\epsilon_1, \epsilon_2$*.*

*More formally, using the partitioning of* $\mathbb{R}^n$*, we can split the integral of Eq.* (3) *into*

$$p_{Y|S=1}(y) = \int_{\mathbf{x}\in\mathbb{R}^p_{\in}} p_{Y|S=1,\mathbf{x}}(y,\mathbf{x}) \frac{p_{S=1|\mathbf{x}}(\mathbf{x})p_{\mathbf{x}}(\mathbf{x})}{\mathbb{P}(S=1)} dx + \int_{\mathbf{x}\in\mathbb{R}^p_{\notin}} p_{Y|S=1,\mathbf{x}}(y,\mathbf{x}) \frac{p_{S=1|\mathbf{x}}(\mathbf{x})p_{\mathbf{x}}(\mathbf{x})}{\mathbb{P}(S=1)} dx \,,$$

*with the goal to upper bound (and hence ignore) the second term, which we consider as an error. This second term can be now bounded as*

$$\int_{\mathbf{x}\in\mathbb{R}^p_{\notin}} p_{Y|S=1,\mathbf{x}}(y,\mathbf{x}) \frac{p_{S=1|\mathbf{x}}(\mathbf{x})p_{\mathbf{x}}(\mathbf{x})}{\mathbb{P}(S=1)} d\mathbf{x} \leq$$

$$\frac{1}{\mathbb{P}(S=1)} \int_{\mathbf{x}\in\mathbb{R}^p_{\notin}} C_Y p_{S=1|\mathbf{x}}(\mathbf{x})p_{\mathbf{x}}(\mathbf{x}) d\mathbf{x} \leq$$

$$\frac{C_Y}{\mathbb{P}(S=1)} \left[ \int_{\mathbf{x}\in\mathbb{R}^p_{\notin}\setminus\mathbb{R}^p_{\notin\subset}} \underbrace{p_{S=1|\mathbf{x}}(\mathbf{x})}_{\leq 1} p_{\mathbf{x}}(\mathbf{x}) d\mathbf{x} + \int_{\mathbf{x}\in\mathbb{R}^p_{\notin\subset}} p_{S=1|\mathbf{x}}(\mathbf{x}) \underbrace{p_{\mathbf{x}}(\mathbf{x})}_{\leq C_{\mathbf{x}}} d\mathbf{x} \right] \leq$$

$$\frac{C_Y}{\mathbb{P}(S=1)} \left[ \int_{\mathbf{x}\in\mathbb{R}^p_{\notin}\setminus\mathbb{R}^p_{\notin\subset}} p_{\mathbf{x}}(\mathbf{x}) d\mathbf{x} + C_{\mathbf{X}} \int_{\mathbf{x}\in\mathbb{R}^p_{\notin\subset}} p_{S=1|\mathbf{x}}(\mathbf{x}) d\mathbf{x} \right] \leq$$

$$\frac{C_Y(\epsilon_2 + C_{\mathbf{X}}\epsilon_1)}{\mathbb{P}(S=1)} \,,$$

*where* $p_{S=1|\mathbf{x}} \leq 1$ *since S is a discrete random variable.*

*We argue about the second part by claiming that both bounds* $\epsilon_1$ *and* $\epsilon_2$ *vanish during learning. Indeed, the form* $s(\mathbf{x}) \to p_{S=1|\mathbf{x}}(\mathbf{x})$ *satisfies the assumption of Eq.* (6) *for a steep enough temperature parameter t, while it is also learning the correct domain* $\mathbb{R}^p_{\notin}$*, so that indeed the assumption of Eq.* (7) *is satisfied, both with inexorably diminishing bounds* $\epsilon_1$ *and* $\epsilon_2$*, respectively.*    $\square$

## B    PROOF OF ASYMPTOTIC CORRECTNESS OF SOFT-BINNING

**Proof:**    *Consider a real value* $x_i \in \mathbb{R}$ *and M sorted bin thresholds* $\beta_{i,j} \in \mathbb{R}$*, i.e.* $\beta_{i,j} < \beta_{i,j+1}$*. From the thresholds* $\beta_{i,j}$*, we construct the bias vector* $b_i \in \mathbb{R}^{M+1}$ *defined as*

$$b_i = (0, \sum_{j=1}^{1} -\beta_{i,j}, \dots, \sum_{j=1}^{M} -\beta_{i,j})^T \,.$$

*Additionally, we define a weight vector* $w \in \mathbb{R}^{M+1}$ *with* $w_j = j$*, so that*

$$w = (1, 2, \dots, M+1)^T \,.$$

*The soft-binning result $z \in [0,1]^{M+1}$ is defined as*

$$z = softmax\left((wx_i + b_i)/t\right) \; .$$

*Now, let $x_i$ be in the $l$-th bin, i.e. $\beta_{i,l-1} < x_i < \beta_{i,l}$, then we now firstly prove that $\forall j \neq l : z_l > z_j$. We do this by showing that the $l$-th logit $\bar{z}_l = w_l x_i + b_{i,l}$ is the largest and hence also has the highest softmax activation.*

*Firstly, note that the bin thresholds are sorted in order, so that for $j < l$ it also holds that $\beta_{i,j} < \beta_{i,l}$. $\bar{z}_l$ is defined as*

$$\bar{z}_l = w_l x_i + b_{i,l} = w_l x_i - \sum_{k=1}^{l-1} \beta_{i,k} \; .$$

*We can simply transform $\bar{z}_l$ into $\bar{z}_{l-1}$ by subtracting $x_i - \beta_{i,l-1}$, so that*

$$\bar{z}_l - x_i + \beta_{i,l} = w_{l-1} x_i - \sum_{k=1}^{l-2} \beta_{i,k} = \bar{z}_{l-1} \; .$$

*Now, as $x_i$ is in the $l$-th bin, we know that $\beta_{i,l-1} < x_i$ and hence $x_i - \beta_{i,l} < 0$. For all other $j < l$ $\beta_{i,j} < x_i$ holds, and hence also $\bar{z}_l > \bar{z}_j$.*

*Now consider the case where $j > l$. Here, it holds that*

$$\bar{z}_l + x_i - \beta_{i,l+1} = w_{l+1} x_i - \sum_{k=1}^{l+1} \beta_{i,k} = \bar{z}_{l+1} \; .$$

*In general, we may transform $\bar{z}_l$ into $\bar{z}_j$ by repeatedly adding $x_i - \beta_{i,k}$ for $k \in [l+1, \ldots, j]$. For all thresholds $x_i < \beta_{i,k}$ holds. Thus, each time we add a strictly negative number to the logit $\bar{z}_l$, which proves that also here $\forall j > l : \bar{z}_l > \bar{z}_j$. Thus, it holds that $\forall j \neq l : \bar{z}_l > \bar{z}_j$*

*Lastly, it remains to prove that with temperature $t \to 0$, $z$ is a one-hot bin encoding, i.e. $z_l = 1$ and $\forall j \neq l : z_j = 0$. The soft-binning of $z_l$ is defined as*

$$\lim_{t \to 0} z_l = \lim_{t \to 0} \frac{\exp(\bar{z}_l/t)}{\sum_{j=1}^{M+1} \exp(\bar{z}_j/t)} = \lim_{t \to 0} \frac{1}{\sum_{j=1}^{M+1} \exp\left((\bar{z}_j - \bar{z}_l)/t\right)} \; .$$

*For $j = l$, the sum term evaluates to $\exp(\bar{z}_l - \bar{z}_l)/t = \exp(0) = 1$. For $j \neq l$, it holds that $\bar{z}_l > \bar{z}_j$ as show previously, and hence in the limit*

$$\lim_{t \to 0} \exp(\bar{z}_j - \bar{z}_j)/t = \exp(-\infty) = 0 \; .$$

*Thus, in the limit $t \to 0$, the denominator sums up to 1 and hence $z_l = 1$, and as the softmax is positive and sums up to zero, it follows that $\forall j \neq l : z_j = 0$.* □

## C  LEARNING MULTIPLE NON-REDUNDANT RULES

The final amendment to our objective comes from the desire to discover multiple exceptional subgroups. In fact, it is often useful to not only look for just a single subgroup, but instead provide a collection of them. For instance, there can be multiple interesting demographic groups in the population under study, each one describing a different sub-population of interest. However, independently running our algorithm multiple times, even with a different initialisation, may lead to discovering either the very same sub-population as before, or even a very similar one.

In other words, running our algorithm multiple times independently could result in a set of subgroups that are redundant. Instead, to achieve a redundancy-free set of results we incrementally run our algorithm multiple times, during each of which we take into consideration all previously found results. That is, assuming at each invocation that we have previously discovered the set of $K$ subgroups $\{S_1, \ldots, S_k\}$, we steer the current result away from the distribution of each of them by imposing an appropriate regulariser. In particular, using once again our adopted measure of statistical dissimilarity, we regularise our objective with the sum of Kullback-Leibler divergences between any previous subgroups and the current candidate. Thus, the amended regularised objective becomes

$$D_{\mathrm{WKL}}\left(p_{Y|S=1} \| p_Y\right) + \lambda \frac{1}{K} \sum_{i=1}^{K} \hat{D}_{\mathrm{KL}}\left(p_{Y|S=1} \| p_{Y|S_i}\right) \; .$$

Figure 5: Subgroup discovery on MNIST handwritten digit dataset. We search the distribution of images $Y \in [0, 255]^{14 \times 14}$ using t-SNE features $X$. SYFLOW clearly recovers the digits 0 and 1.

## D  SUBGROUP DISCOVERY ON IMAGE DATA

We consider the setting with a multi-variate target variable $Y$. Our probabilistic framework seamlessly accommodates such a setting. To assess the scalability of SYFLOW in this regard, we utilize the well known MNIST handwritten digits image dataset (LeCun et al., 2010). In this experiment, we investigate the distribution of images for the digits zero and one, which we treat as the target variable $Y \in [0, 255]^{14 \times 14}$. Here, we use the normalising flows to model the distribution of digits, represented as 196-dimensional vectors.

As descriptive features $X$, generate correlated two-dimensional features using t-SNE (Van der Maaten & Hinton, 2008). This allows us to visually evaluate the discovered subgroups by sampling the discovered subgroup distribution, as well as plot the rule as a box in the t-SNE feature space. SYFLOW now searches the feature space $X$ that thanks to t-SNE contains two distinct clusters for each digit. We show the two discovered subgroup rules $s(X; \alpha, \beta, a, t)$ and distributions $P(Y|S = 1)$ in Figure 5. The black boxes represent the thresholds learned by SYFLOW, and show that SYFLOW ably recovers the subgroups each belonging zero and one respectively. On the left, we show five samples of the learned distributions $P(Y|S_0 = 1)$ and $P(Y|S_1 = 1)$. They too resemble the digits zero and one closely, demonstrating that the combination of learning the generating distribution and simultaneously fitting a rule works.

## E  ALGORITHM AND HYPERPARAMETERS

In this section, we provide pseudocode and detail the hyper parameters used in the experiments.

---

**Algorithm 1:** SYFLOW($X, Y, \lambda, prior\_sg\_flows, n\_epochs\_1, n\_epochs\_2$)

---

1  **for** $i \leftarrow 1$ **to** $n\_epochs\_1$ **do**
2  $\quad$ minimize $-p_{\mathcal{Y}}$.log-likelihood($Y$);
3  **for** $i \leftarrow 1$ **to** $n\_epochs\_2$ **do**
4  $\quad$ $subgroup\_label \leftarrow$ rule_learner($X$);
5  $\quad$ $log\_p \leftarrow$ flow_Y.log_p($Y$);
6  $\quad$ $log\_p\_sg \leftarrow$ flow_Y_sg.log_p($Y$);
7  $\quad$ $kl\_divergence \leftarrow subgroup\_label \cdot (log\_p\_sg - log\_p)$;
8  $\quad$ $regularization \leftarrow$ zeros(len($Y$));
9  $\quad$ **for** $sg\_flow$ **in** $prior\_sg\_flows$ **do**
10 $\quad\quad$ $log\_p\_sg\_2 \leftarrow sg\_flow$.log_p($Y$);
11 $\quad\quad$ $regularization \leftarrow regularization + (log\_p\_sg - log\_p\_sg\_2)$;
12 $\quad$ $loss \leftarrow -kl\_divergence - \lambda \cdot regularization$;
13 $\quad$ $loss$.backwards();
14 $\quad$ $classifier$.step();
15 $\quad$ update $flow\_Y\_sg$;

---

| Dataset | $t$ | $\gamma$ | $lr_{Flow}$ | $lr_\pi$ | $epochs_{Flow_Y}$ | $epochs_{Flow_{Y_s}}$ |
|---|---|---|---|---|---|---|
| 10 | 0.1 | 0.6 | $5 \times 10^{-2}$ | $5 \times 10^{-2}$ | 1000 | 500 |
| 50 | 0.1 | 0.6 | $5 \times 10^{-2}$ | $5 \times 10^{-2}$ | 1000 | 500 |
| 100 | 0.1 | 0.6 | $5 \times 10^{-2}$ | $5 \times 10^{-2}$ | 1000 | 500 |
| 250 | 0.1 | 0.6 | $5 \times 10^{-2}$ | $5 \times 10^{-2}$ | 1000 | 1500 |
| 500 | 0.1 | 0.6 | $5 \times 10^{-2}$ | $5 \times 10^{-2}$ | 1000 | 1500 |
| 750 | 0.1 | 0.8 | $5 \times 10^{-2}$ | $5 \times 10^{-2}$ | 1000 | 1500 |
| 1000 | 0.1 | 0.6 | $5 \times 10^{-2}$ | $5 \times 10^{-2}$ | 1000 | 1500 |

Table 1: Hyperparameters of SYFLOW for the **Scalability** experiments.

| Dataset | $t$ | $\gamma$ | $lr_{Flow}$ | $lr_\pi$ | $epochs_{F_Y}$ | $epochs_{Flow_{Y_s}}$ |
|---|---|---|---|---|---|---|
| wine | 0.2 | 1 | $5 \times 10^{-2}$ | $5 \times 10^{-2}$ | 2000 | 5000 |
| insurance | 0.2 | 1 | $5 \times 10^{-2}$ | $1 \times 10^{-2}$ | 5000 | 2000 |
| wages | 0.2 | 1 | $5 \times 10^{-2}$ | $1 \times 10^{-2}$ | 5000 | 2000 |
| mpg | 0.2 | 1 | $1 \times 10^{-2}$ | $1 \times 10^{-2}$ | 5000 | 2000 |
| life | 0.2 | 1 | $1 \times 10^{-2}$ | $1 \times 10^{-2}$ | 5000 | 2000 |
| bike | 0.2 | 1 | $1 \times 10^{-2}$ | $1 \times 10^{-2}$ | 2000 | 5000 |

Table 2: Hyperparameters of SYFLOW for the real world experiments in Section 5.2.

## F  HYPERPARAMETERS FOR EXPERIMENTAL EVALUATION

### F.1  SYNTHETIC EXPERIMENTS

**Cutpoints**  We used for SYFLOW the following the following hyperparameters: $t = 0.1$, $\gamma = 0.8$, $lr_{Flow} = 5 \times 10^{-2}$, $lr_\pi = 1 \times 10^{-2}$, $epochs_{Flow_Y} = 1000$ and $epochs_{Flow_{Y_s}} = 500$.

**Complex Target Distributions & Scalability**  For SD-$\mu$, SD-KL and RSD we used the same hyperparameters for both settings. Since PRIMPhas no hyperparameters, no tuning is required. For SD-$\mu$ and SD-KL, we used 10 cutpoints, a beam width of 100 and maximal exploration depth of 10. Due to runtime issues of RSD we reduced the number of cutpoints to 2 and the beam width to 50. For SYFLOW, we used in the experiment for different target distributions, the following hyperparameters: $t = 0.1$, $\gamma = 0.6$, $lr_{Flow} = 5 \times 10^{-2}$, $lr_\pi = 1 \times 10^{-2}$, $epochs_{Flow_Y} = 1500$ and $epochs_{Flow_{Y_s}} = 1000$. The hyperparameters for the scalability experiment are shown in Table 1.

### F.2  REAL WORLD DATA & CHARACTERIZING CLASSIFICATION ERRORS

The hyperparameters for the real world experiments conducted in Section 5.2 data can be seen in Table 2. For the classification errors we used the hyperparameters: $t = 0.2$, $\gamma = 0.25$, $lr_F = 5 \times 10^{-2}$, $lr_\pi = 1 \times 10^{-2}$, $epochs_{F_Y} = 1500$ and $epochs_{F_{Y_s}} = 500$.

## G  LIMITATIONS

SYFLOW is a general framework for subgroup discovery, which can be applied to any domain with a differentiable likelihood. However, there are some limitations to our approach. As of now, our learned predicates are restricted to a single interval per feature and hence lack expressive power for scenarios where multiple intervals are required. Rules which are not based on logical conjunctions of predicates are not supported yet, and interesting to explore in future work. As a neural network based approach, SYFLOW has a generally higher sample complexity than a classical, mean statistic based approach as it needs to learn the distribution of the target variable. And finally, the optimisation process is based on gradient descent so that we can not give any formal guarantee on reaching a global optimum of the objective, as is the case for most recent machine learning approaches.

| Dataset | Method | Rule |
|---|---|---|
| wine | SYFLOW | $0.21 <$ volatile acidity $< 1.28 \wedge 7.87 <$ alcohol $< 10.70$ |
| | RSD | volatile acidity $\geq 0.3 \wedge$ alcohol $< 9.7 \wedge 41.0 \leq$ free sulfur dioxide $< 51.0$ $\wedge 6.5 \leq$ fixed acidity $< 7.6 \wedge$ density $\geq 1.0$ |
| | SD-KL | alcohol $< 12.4$ |
| | SD-$\mu$ | alcohol $\geq 10.4 \wedge$ free sulfur dioxide $\geq 15.1 \wedge$ total sulfur dioxide $< 194.9 \wedge$ density $< 1.0$ |
| insurance | SYFLOW | smoker $= 0$ |
| | RSD | bmi $\geq 30.0 \wedge$ smoker $= 1 \wedge$ age $\geq 39.0$ |
| | SD-KL | age $< 59.0 \wedge$ smoker $= 0$ |
| | SD-$\mu$ | smoker $= 1$ |
| wages | SYFLOW | sex $=$ female $\wedge 2.68 <$ education $< 15.42$ |
| | RSD | $64.3 \leq$ height $< 66.05 \wedge$ education $< 12.0 \wedge 42.0 \leq$ age $< 63.0$ |
| | SD-KL | education $< 17.0$ |
| | SD-$\mu$ | height $\geq 63.1 \wedge$ sex $=$ male $\wedge$ ed $\geq 11 \wedge$ age $\geq 27$ |
| mpg | SYFLOW | $5.76 <$ cylinders $< 9.92 \wedge 187.26 <$ displacement $< 587.84$ |
| | RSD | weight $\geq 3947.5$ |
| | SD-KL | weight $\geq 2809.7 \wedge$ model-year $< 80$ |
| | SD-$\mu$ | cylinders $= 4.0 \wedge$ weight $< 2809.7$ |
| life | SYFLOW | $0.31 <$ Income composition of resources $< 1.14$ $\wedge 0 <$ Adult Mortality $< 181.78 \wedge 11.82 <$ Schooling $< 23.99$ |
| | RSD | Income composition of resources $\geq 0.797 \wedge$ Year $< 2003.0$ |
| | SD-KL | HIV/AIDS $< 4.5 \wedge$ Income composition of resources $< 0.84$ $\wedge$ Income composition of resources $\geq 0.56 \wedge$ Adult Mortality $< 253.4$ |
| | SD-$\mu$ | HIV/AIDS $< 4.5 \wedge$ Income composition of resources $\geq 0.56$ $\wedge$ Adult Mortality $< 212.0$ |
| bike | SYFLOW | $1.64 <$ temp $< 17.22$ |
| | RSD | season $< 3.0 \wedge$ atemp $< 19$ |
| | SD-KL | temp $\geq 10.66 \wedge$ atemp $\geq 13$ |
| | SD-$\mu$ | mnth $\geq 3 \wedge$ temp $\geq 17.63 \wedge$ hum $< 0.82$ |

Table 3: Symbolic subgroup descriptions for real life datasets in Section 5.2

## H SUBGROUP DESCRIPTIONS

We show in Table 3 examples of subgroups found on the real life datasets. For each method we select the subgroup with the highest size-corrected KL. Since PRIMP did not find relevant subgroups for most datasets, we do not consider it in the table.

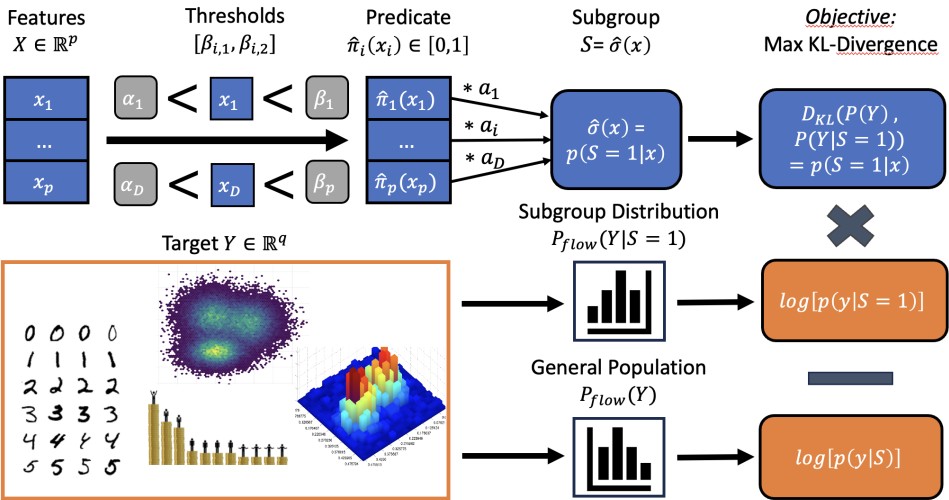

Figure 6: Overview over architecture of SYFLOW. Each feature $x_i$ is thresholded with learned bounds $\alpha_i$ and $\beta_i$ and combined into a rule with weights $a_i$ (top). In parallel, we estimate the general likelihood $p_Y(y)$ and in the subgroup $p_{Y|S=1}(y)$ using normalising flows (bottom) and aggregate them into the KL-Divergence (right).

