# OpenReview forum: "Characterizing Exceptional Distributions with Neural Rule Extraction"
_ICLR.cc/2024/Conference — Submitted to ICLR 2024_

### Official Review · Reviewer_fxuS · 2023-10-18

**Soundness:** 2 fair
**Presentation:** 2 fair
**Contribution:** 3 good
**Rating:** 3
**Confidence:** 4

**Summary:**

The authors propose a method, SYFLOW, for discovering subgroups of a dataset where the distribution of a target variable deviates significantly from its distribution in the population as a whole. Their method utilizes normalizing flows to approximate the distribution of the target variable, and therefore isn't limited to detecting only distributions of specific parametric types, like existing previous methods. SYFLOW also uses an end-to-end differentiable quality score, so the thresholds used to define the subgroup do not need to be pre-defined discretizations of the features, nor is there a need for computationally expensive combinatorial optimization procedures. Experiments on synthetic and real data show that in terms of the deviation of the discovered subgroup from the population, their method achieves improved performance over competitor methods, or similar performance with vastly reduced runtime.

**Strengths:**

The problem under consideration is important and has widespread practical utility. As the authors mentioned, existing methods for subgroup discovery usually require pre-specified discretizations of continuous variables, and in such cases either suffer from intractable runtime or poor accuracy. Using an end-to-end differentiable learning framework to this problem is a novel and exciting application of ML techniques, and using normalizing flows to deal with arbitrary target distributions while preserving useful gradients is a clever solution to a tricky problem.

Some of the empirical results are also promising. SYFLOW obtains large gains in either performance or runtime compared to previous baselines, and it appears to be more widely applicable across difference problems, as compared to other methods which may have excellent performance on certain problems (e.g. where some of their distributional assumptions about the target variable are justified) but suffer greatly when their assumptions aren't met. In contrast, SYFLOW's performance is consistently high.

**Weaknesses:**

My main concern is with the technical clarity/correctness of Section 3.3. This is significant because this section provides a mathematical justification for the proposed method, but there are a number of mistakes, undefined terminology, and unclear logical reasoning which cast serious doubt on the theoretical soundness.

**Undefined or unclear terminology.** Many of the quantities used in the derivation are not defined. The quantities $p_Y$, $p_{Y|S=1}$, $p_{Y|S=1, \mathbf{x}}$, etc. are presumably densities and conditional densities of these random variables with respect to the Lebesgue measure, but these are never defined explicitly. They are sometimes used interchangeably with the probability laws themselves, e.g., the authors also refer to $D_{\mathrm{KL}}(p_{Y|S=1}\|p_Y)$.

Most confusing are the random variable $S$ and related quantities. What, precisely, is the definition of the random variable $S$? Based on the initial motivation, it seems that $S(\mathbf{x}) = \sigma(\mathbf{x})$ is just the indicator of whether or not a point belongs to the correct subgroup. But my initial understanding was that the subgroup is a deterministic quantity, in which case $p_{S=1|\mathbf{x}}(\mathbf{x})$ is either 0 or 1 and is in fact equal to $S(\mathbf{x})$. If this is incorrect, what is the underlying probabilistic process? It seems like the authors may be confusing a soft approximation for a deterministic quantity with a true probability. If there is some true underlying probabilistic mechanism for $S$, it should be defined explicitly. If there is not an underlying probabilistic mechanism, then most of the derivation in Section 3.3 does not make sense.

**Typos.** There are several technical typos which make the paper harder to understand, including:
1. Pg. 3, definition of $\mathcal{M}(x)$: The numerator should be $p$, not $D$.
2. Pg. 3, definition of $s(x;\alpha,\beta,a,t)$: The summand in the denominator should be $a_i\hat{\pi}(x_i;\alpha_i,\beta_i,t)^{-1}$.
3. Pg. 4, equation (2): I think this should be $D_{\mathrm{KL}}(p_{Y|S=1}\| p_Y)$.
4. Pg. 4, equation (5): Based on the motivation, I think equation (5) should be the integral over $\mathbb{R}^p_{\not\in \subset}$. It's also not clear why $\mathbb{R}^p_{\not\in}$ is being broken up into two regions, since the intuitive justification that $\epsilon_1$ and $\epsilon_2$ both approach 0 could just be directly applied to the whole region $\mathbb{R}^p_{\not\in}$.
5. Pg. 4, last inequality: The first integral also has a factor of $p_{\mathbf{x}}(\mathbf{x})$, so I believe this should also have a factor of $C_{\mathbf{X}}$ in the bound.
5. Pg. 5, equation (7): It seems the approximation is being used in the first equality, not the second.

**Experimental results.** It is suspicious that the number of features which actually need to be thresholded $f_p$ is so small compared to the ambient dimension. What happens when a more significant fraction of the features need to be thresholded? Also, I assume the information about which features must be thresholded is not being supplied to the algorithm, but this isn't stated explicitly.

For the real-world data experiments, reporting the size-corrected KL divergence is basically reporting the value of the objective that SYFLOW optimizes, so it is unsurprising that SYFLOW has the best performance. Including some other metrics like deviations of common statistics (mean, standard deviation, etc.) from their population levels would be a more convincing comparison.

In Fig. 2(c), the results of SYFLOW look fairly unstable. Can the authors comment on this?

In Figs. 3(b) and 3(c), it is not obvious that the distribution of the y-values for the blue points is significantly different from the distribution for the overall population.

Almost none of the results have any measures of uncertainty/error bars. The authors should run multiple trials and report the variability of the methods; at present, it's unclear if the differences in performance are statistically significant.

The authors also don't describe or give explicit reference to the baseline methods.

**Minor: Missing references.** Lastly, there are also some missing references that should be discussed, including:
1. Christopher Sutton, Mario Boley, Luca M Ghiringhelli, Matthias Rupp, Jilles Vreeken, and Matthias Scheffler. Identifying domains of applicability of machine learning models for materials science. Nature communications, 11 (1):1–9, 2020. (Relevant for the COVID example.)
2. Zachary Izzo, Ruishan Liu, and James Zou. Data-driven subgroup identification for linear regression. ICML, 2023. (Another method which does subgroup discovery without pre-specified discretizations.)
3. What is the citation for the "most related work of its kind" described in the last paragraph of Section 4.1?

**Questions:**

1. What is the precise definition of the random variable $S$?

2. None of the functions in Algorithm 1 (Appendix C) are defined. Can the authors provide a complete, precise definition of their algorithm? It's especially unclear to me how normalizing flows will be used to learn $p_{Y|S=1}$ in a way that keeps nonzero gradients with respect to the learned boundaries. Also, what is the regularization term being used in the algorithm?

3. How does the method handle discrete features, as it does in the COVID-19 experiment?

---

> ### Author Response · Authors · 2023-11-22
>
> We appreciate the detailed feedback and the kind suggestions.
>
> ## I. Density notation
> In our work we do introduce the used quantities as densities, but leave implicit the measure-theoretic formalisation as it is not necessary for the needs of this work. As for the divergences, indeed, previously we slightly abused notation to avoid the introduction of the laws, again, as these are not used elsewhere in the manuscript. To avoid confusion, however, we now introduce this notation in the updated manuscript at the end of the preliminary section (Sec 2), so that we can now adopt the usual notation.
> ## II. Defining S
> The latent r.v. S was introduced in the manuscript in the last part of sec. 2. as a (latent) binary r.v. that represents the membership in a subgroup. However, we agree that it could have been missed or been too terse. To rectify this problem we clarify its role in our model, as well as the underlying model in our inference in the beginning of section 3.1, in addition to the preliminaries.
> ## III. Typos
> We thank the reviewer for the notes. We have incorporated them all in the revised manuscript.
> 4. 5. We believe our original derivation is correct. However, we acknowledge that the original writeup could be improved. We therefore moved the derivation to the appendix, and, given the added space, we explain the two mechanisms that need different bounds, and also demonstrate why the former bound is not depending on $C_X$.
> ## IV. Experimental Results
> 1. Number of features in synthetic data
> There are two main reasons behind the maximum of 10 features which define the subgroup. Firstly, the premise of our work (and subgroup discovery in general) is that the results have to be announced to a human, and therefore fewer features lead to more succinct descriptions and are therefore considered superior, from the interpretability perspective.
> The number of 10 was chosen as, intuitively, we considered it as a round number that was already past the limit of what could reasonably be interpreted by a human expert (e.g., doctor).
> Secondly, increasing the amount of features for thresholding also comes with a data generation problem. Imposing a rule with 100 conditions on independent features would lead to an interval that covers the entire feature domain ($0.2^(1/100)≈1$), or if the features are correlated, extreme redundancy in which feature to actually threshold. This would allow a different process to start becoming apparent, in which arbitrary features would simply become relevant simply by random chance, and necessitate additional mechanisms for testing (e.g., statistical corrections for random correlation).
> 2. KL vs other metrics
> We agree with the reviewer that comparing to methods that optimise different objectives is inherently difficult. However, KL divergence is a more general measure of distribution dissimilarity than other widely used criteria; for instance, when assuming Gaussian target distributions, then there is a monotone bijection between KL and difference of means, while in other cases we could easily craft a case where the KL divergence exceeds an arbitrary bound, while the mean shift remains close to zero.
> In other words, a low mean difference does not imply that the resulting subgroup is not interesting, as in the exponential wage distribution example from Fig. 3b.
> 2. Stability of results
> The issue came primarily from a small number of iterations on similar datasets.
> We increased the number of iterations and used a newly generated randomised dataset for each of them to now improve the smoothness of the experimental measurements.
> 3. Description of baselines
> We fixed this in the revised manuscript.
> ## V. References
> 1. , 2. We have added the suggested references to the introduction.
> 3. That would be the Proença et al. We rephrased the relevant sentence to make clear that we were still referring to the last mentioned method.
>
> ## VI. Questions
> 1. Please see II. This has been incorporated in the revised manuscript in preliminaries (last sentence) and Sec. 3.1 (first sentence).
> 2. Algorithms
> Briefly, we first fit a normalising flow to the entire data, then initialise $p_{Y|S=1}$ to the former model, and perform SGD steps that jointly update both the conditional target model and the membership rule. To avoid the gradient of the rules from vanishing, we follow a simulated annealing scheme, by starting with a very moderate temperature parameter.
>
> 3. Although our model is primarily aimed at real features, ordinal and binary features can also easily be incorporated, by simply treating the values as real. In our experiments we treat the features as real numbers, resulting in logistic-like regression. For discrete data we resort to one-hot encoding, where we create a binary variable that indicates the presence/absence of each discrete value. This allows SYFLOW to easily select individual values for its rules and matches the ability of standard subgroup discovery methods for dealing with discrete data.

---

> ### Comment · Reviewer_fxuS · 2023-11-28
>
> Thanks to the authors for their response. Some improvements have been made in the revised manuscript, but I will maintain my score for the following reasons:
>
> **Math:** There are still many technical typos which make the results difficult to understand. Some examples:
> 1. (First paragraph of Section 3.1) "That is, from the statistical inference perspective, we assume that $S|\mathbf{X} = \mathbf{x} \sim \textrm{Bernoulli}(\sigma(\mathbf{x}; \alpha, \beta))$. The Bernoulli parameter should be $s$, not $\sigma$, since $\sigma$ is the 0-1 (hard) subgroup rule. (If $\sigma$ is correct, then $S = \sigma(\mathbf{x}; \alpha, \beta)$ with probability 1.)
> 2. (Theorem 2) The $d\mathbf{x}$ was placed on the wrong side of the inequality, instead of with the integral.
> 3. $S$ is not a single random variable, but rather a random function that depends on $\mathbf{x}$.
>
> It's possible that these problems can be resolved with just notational fixes and not by changing the fundamental content, but it is not the reader's job to do so. The paper should be carefully edited so that all mathematical statements are precise and correct before it is published.
>
> **Experiments:** There are still no error bars on most of the results, so the significance is still unclear. Regarding the choice to report KL divergence vs difference in some common statistics, I agree with the authors that one can construct examples where these statistics will not differ even though the distributions are quite different. I would encourage you to report these results (or perhaps some other metrics quantifying difference between probability distributions, e.g. Wasserstein distance) in *addition* to the KL divergence to show that your method has some amount of robustness to the measure of dissimilarity.
>
> I believe the method is promising and I encourage the authors to resubmit after making these changes.

---

### Official Review · Reviewer_MKfq · 2023-10-27

**Soundness:** 2 fair
**Presentation:** 2 fair
**Contribution:** 2 fair
**Rating:** 5
**Confidence:** 4

**Summary:**

This paper proposes a method for extracting interpretable rules that describe regions in which the distribution of $Y$ differs within the joint distribution $(X, Y)$.
Traditionally, this problem of subgroup discovery has been approached using combinatorial methods.
In this study, the authors proposeed solving this problem as a differentiable problem.
Specifically, the authors approximate the rules defining the regions of $X$ using differentiable soft-binning functions and estimate the distribution of $Y$ using Normalizing Flow.
The objective function is defined as the KL divergence between the distribution of $Y$ and the distribution of $Y$ within the subgroup.
Finally, the subgroup is estimated by miximizing this KL divergence using gradient-based methods.
In the experiments, the authors reported that the proposed method achieved comparable or superior performance with shorter computation times compared to existing methods.

**Strengths:**

The strength of this paper is the performance of the proposed method.
In particular, its superior performance and shorter computation times would be a significant practical advantage.

**Originality, Quality**

An original aspect of the proposed method is its capability to handle multi-dimensional outputs for $Y$ by using Normalizing Flow.

**Clarity**

There appears to be considerable room for improvement in terms of the clarity of this paper. For more details, please refer to the "Weaknesses" below.

**Significance**

The superior performance and shorter computation times of the proposed method would be a significant practical advantage.

**Weaknesses:**

The weaknesses of the paper include "Ambiguity in Problem Definition," "Lack of Explanation for Existing Methods," and "Insufficient Explanation of Experimental Setup."

**Ambiguity in Problem Definition**

The paper uses a latent variable $S$ to represent subgroups without providing a clear definition of it.
An unclear aspect of this problem definition is how the latent variable $S$ is determined.
While the paper treats the latent variable $S$ as a variable related to $X$ and $Y$, the assumption regarding this relationship is not explicitly stated.
However, estimating the latent variable without assumptions is evidently impossible.
Consequently, it is possible that there are implicit assumptions in the problem definition of this paper that have not been explicitly mentioned.

Also, in the paper, functions such as $s$ and $\sigma$ are introduced to estimate the undefined $S$.
However, it appears that the proposed method is estimating not the true undefined $S$ but simply $\sigma$ that maximizes (8).
This seems to diverge from the problem definition that assumes the existence of the true latent variable $S$.

**Lack of Explanation for Existing Methods**

In the experiments, several methods such as SD-$\mu$, SD-KL, RSD, and BumpHunting are introduced as the baseline methods to be compared with.
However, these baseline methods are introduced without references, making it impossible to determine what these methods do and whether they are valid as baseline methods.
Additionally, the absence of supplemental code makes the results irreproducible.

**Insufficient Explanation of Experimental Setup**

Details of the experimental setup for the proposed method are not provided in the paper.
For instance, how the temperature parameter $t$ was set and decayed, and how the size-corrected KL's $\gamma$ was configured, are not described.
These parameters are crucial for achieving favorable results with the proposed method, both in this experiment and for practical applications.
Describing the methods for setting these parameters, as well as other settings such as optimization algorithms, their parameters, and the number of training epochs, is essential to ensure the reproducibility of the results.

**Questions:**

* What is the definition of the latent varianbel $S$? How it is related to $X$ and $Y$?
* What are SD-$\mu$, SD-KL, RSD, and BumpHunting? Please provide their reference, and also discuss how and why they are relevant as the baseline methods to be compared with.
* Please describe the details of the experimental setup for the proposed method. For instance, how the temperature parameter $t$ was set and decayed, and how the size-corrected KL's $\gamma$ was configured? What optimization algorithms are used, and how their parameters and the number of training epochs are determined?.

---

> ### Author Response · Authors · 2023-11-22
>
> Thank you for your detailed feedback.
> 1. The latent random variable S was introduced in the manuscript in the last part of Sec. 2. as a (latent) binary random variable representing the membership of a sample to the subgroup. We see that this is both terse and easy to miss. We therefore now discuss its role in the model and during inference in more detail in the beginning of Section 3.1 of the updated manuscript.
> 2. We now give an introduction to describe the baselines in the beginning of the experiment section.
> 3. We now describe the hyperparameters, data generation and training procedure in the Appendix of the revised manuscript.

---

### Official Review · Reviewer_mMzX · 2023-10-31

**Soundness:** 3 good
**Presentation:** 2 fair
**Contribution:** 3 good
**Rating:** 6
**Confidence:** 3

**Summary:**

The authors introduce Syflow that contains components to the task of learning distributions over subgroups in a data set, particularly with an emphasis on interpretability and scalability. It combines the use of normalising flows for flexibility in modelling arbitrary distributions with a differentiable objective function for optimisation. The authors perform experiments to evaluate Syflow on a number of datasets and compare it to baseline methods.

**Strengths:**

- The combination of normalising flows, differentiable KL-divergence objectives and learnable feature bounds for subgroup discovery their integration for this specific application offers a new approach.
- Importance and Applications: The problem tackled is particularly relevant for healthcare, aiming to identify interpretable subgroups in medical data. The approach has the potential to inform clinical decision-making and has high utility in healthcare applications.
- Relevance to ICLR Community: The method encompasses topics of interest to the ICLR community, including normalising flows, differentiable objectives, and scalability. The focus on both theoretical and practical aspects of machine learning makes it a suitable contribution for the conference.

**Weaknesses:**

- The writing could be sharpened in places, namely the introduction. In particular, it is lacking in citations to back up the statements and help guide the reader. The last three paragraphs in the introduction make many statements about previous methods and the state of the field yet there are no references, with the exception of two citations in the first sentence. I believe many of these references appear in the related works section, but it is confusing for the reader to have to cross reference the sections and find the relevant citation.
- This also applies to parts of the derivations, e.g. moving from eq. 2 to eq. 3 using "rules of marginal probability and then that of Bayes" would be better written out more explicitly to help the reader.
- The results are promising but they could be more convincing. There seems to be from single runs which makes it difficult to estimate significance. There should be error bars on the figures and errors in the table would be a very helpful addition.

**Questions:**

- There are two limit parameters two set the range for the subsets, what if the subsets are multi-modal? Are multiple ranges applied to each feature? I.e. to capture both a subset with young adults and another with older individuals?
- Cutting the abstract into three paragraphs makes it easier to read since it is so long but I suggest to the authors that they solve it instead by making the abstract more concise, in one paragraph
- This paper uses the style guide for the 2023 conference, not 2024
- The figure captions could be more descriptive to help the reader understand the context better, also, the captions could be a *little* bit larger
- Figure 3a should be a table and not a figure, also, is it supposed to be "quantitative"?

**Details Of Ethics Concerns:**

Although a specific ethics review is not required, it would be a very welcome addition from the authors to have them add a discussion on the potential ethical concerns. This is particularly important in light of the potential application in health. What are the potential failure modes of the method? Are there any specific groups or types of groups that might be disproportionally affected? What could a practitioner do to mitigate and understand potential biases? The authors are well placed to shed some light on this issue.

---

> ### Author Response · Authors · 2023-11-22
>
> We thank the reviewer for their kind feedback. In regards to the posed questions:
> 1. Currently, a subset is limited to one continuous interval per feature, so that we would need two subsets to capture two ranges right now. Our newly introduced regularizer helps find such contrasting subsets.
> 2. We have added a description of the baselines to the revised manuscript and a discussion on failure modes/limitations in the respective Appendix.
>
> Further writing such as making the abstract more concise and updating the captions are underway and will be incorporated as soon as possible. Thank you very much for your helpful feedback.

---

> > ### Comment · Reviewer_mMzX · 2023-11-23
> >
> > I thank the authors for the response. I will be keeping my score.

---

### Official Review · Reviewer_SrBh · 2023-11-01

**Soundness:** 2 fair
**Presentation:** 3 good
**Contribution:** 2 fair
**Rating:** 3
**Confidence:** 3

**Summary:**

The paper proposed SYFLOW, a method to discover locally optimal subgroups with exceptional distributions from a dataset using neural rule extraction.

**Strengths:**

- Originality: the paper demonstrates creative combinations of existing ideas for subgroup detection
- Clarity:The paper is well-organized with clear description of problem formulation, related work, experiment design and results comparison. Proposed method is evaluated on both synthetic data and public benchmark datasets.

**Weaknesses:**

1. Identification of a single subgroup versus the rest of population might of low practical utility, with comparison to identify multiple subgroups that are different from each other.
2. The proposed method will generate rule-based subgroup characteristics based on one or more individual raw features, which might miss the potential hypotheses such as weighted combination of different features to distinguish and characterize the potential subgroup, e.g. 0.5*Age + I(Heart Failure), etc.
3. Identify distribution-wise separable groups might be meaningful from statistical perspective, but might not be meaningful from real-world view (e.g. rule of ages might be of highest KL-divergence in identification of subgroup A, but the subgroup of the 2nd highest KL based on gender might matter most in clinical perspective). Also, the quantification measure of divergence might result in potential over-fitting on existing data and cannot be generalized to unseen data. How to ensure robustness of the proposed method?
4. Not sure whether experiment results are cross-validated in Figure 3.
5. It's not clear how the characteristics of subgroups identified in different models differ from each other in Figure 3. It will be helpful to review the rules similar to Figure 4, but compared across different models.
6. The paper lacks the description of limitations in final conclusion part.

**Questions:**

1. How to validate the robustness of the proposed approach?
2. Will the proposed approach guarantee best classification if subgroup is known?

---

> ### Author Response · Authors · 2023-11-22
>
> Thank you for your detailed feedback.
> 1. Although it may be interesting to find multiple subgroups, we note that it is not our main objective.  We here focus on developing a method that can: 1) deal with arbitrary target distributions, 2) find symbolic and highly interpretable descriptions of numeric features in a differentiable end-to-end fashion and 3) significantly reduce the runtime compared to state-of-the-art baselines.
> Nonetheless, our method can be extended in a straightforward manner to find a diverse set of subgroups. The main idea is to iteratively search for subgroups and extend the optimization objective with a regularizer of the form $KL(P_{Y|S} || P_{Y|S_{i-1}})$, described in Appendix C. This allows us to find subgroups that differ from the overall distribution and simultaneously differ from previously found subgroups.
> 2. The trade-off between explainability and expressivity is a general problem in the area of interpretability. The rules/descriptions we are interested in, i.e. boolean predicates over binned features, have clear and actionable meaning. The subgroup in Fig 3b, for example, is characterized by the rule “education < 16 AND I(Female)”, permits us to say for any current or future data point whether it is part of the subgroup or not. Rules of the suggested form, e.g. $0.5*$education + I(Female), have a vastly different meaning. This includes the inclarity of what “$0.5*$education” would mean in terms of which samples are in or outside the subgroup, and, the inclarity of what the semantics of adding a real-valued literal to a boolean-valued variable would be. While for classification these aspects perhaps make sense, it does not for descriptive purposes.
> Nonetheless, if such linear rules are of interest, SyFlow can be easily adapted by replacing the rule layer with a linear layer. In general, the rule layer can be replaced by any function that maps features to a probability in a differentiable manner.
> 3. We show in the covid experiment the ability of SyFlow to adjust for factors which may be not clinically relevant, by means of searching the joint distribution of target quantity (e.g. survivial) and factor to adjust for (e.g. age). This advantage is unique to SyFlow, as it is the only method that is designed for non-gaussian multi-variate data.
> 4. They are not. Our goal is description, rather than prediction, and hence cross-validation does not make much sense. After all, the point in description (unsupervised) is to provide insight into the available data, rather than achieving the best possible generalization to unseen data, as is the point in predictive (supervised) settings. To show that our method discovers subgroups with exceptionally distributed target variables, we report the size-corrected KL-divergence of the top subgroup per method
> 5. This is a fair point. We will include the discovered rules in the Appendix. Note, however, that these results only make sense when the discovered subgroups have meaningfully high divergences; for wine and wages, for example, the competing methods all fail to discover a subgroup with an exceptional distribution of the target variable, so it makes little sense to inspect these rules.
> 6. We added a Limitations section to the Appendix, and propose to prune some of the figures in the experiments to make space for this in the main manuscript.
>
> For your questions:
> 1. We test SyFlow across a variety of data distribution types as well as on a diverse set on real world datasets, where it is consistently amongst the top methods.
> 2. We are not exactly sure what you mean by this, but in our experiments SyFlow recovers a planted ground truth subgroup (interpretable as class) accurately.

---

### Official Review · Reviewer_eY6J · 2023-11-05

**Soundness:** 3 good
**Presentation:** 2 fair
**Contribution:** 2 fair
**Rating:** 5
**Confidence:** 4

**Summary:**

The work presents a method for identifying a salient subset given a labeled dataset. This subset is defined by a logical conjunction of an interval over each feature, and it is salient in that the distribution of the label/target differs significantly from that of the population. The paper described a differentiable method of learning both the saliency and the optimal subset (i.e. the feature intervals). Specifically, it defines soft thresholding to approximate each conjunctive clause, and thereafter employs normalizing flows for density estimation for this subset and the population; the gap between these is maximized to identify the optimal subset using SGD. Empirical evaluation over synthetic and real dataset shows the method is effective.

**Strengths:**

The work does a good job of presenting the problem and defines a promising method to solve it.

**Weaknesses:**

- In Section 3.3, the divergence (eq 2) is first approximated using eq 7, and subsequently eq 7 is approximated via sampling. Why couldn't the eq 2 be directly approximated via sampling?
- The evaluations are somewhat inadequate. First, what is SD-\mu in Section 5.1? Second, what are the cutpoints there? Finally, what are the various columns in Table/Figure 3 (i.e. where are these methods defined)?
- The method as presented is primarily for identifying a single salient subset. It would be much more useful if it could be used for multiple salient subsets, that are further distinct from each other (to get diversity).
- The paper mentions on page 2 that `As we show, this quantisation can greatly reduce the quality of the results` -- where is this shown?
- Please cite the relevant literature the first time you use/define terms (e.g. normalizing flows in the abstract).
- In Section 2 "Preliminaries" first para, I suppose in \pi(x;\alpha_i,\beta_i) you meant to use x_i (and not x).

**Questions:**

Please address the weaknesses above.

---

> ### Author Response · Authors · 2023-11-22
>
> Thank you for your detailed feedback.
> 1. Eq (2) does not involve the subgroup rule function s, while Eq (7) does and therewith gives us gradients to optimize the rule s.
> 2. We have added a paragraph describing the baseline methods to the updated manuscript.
> 3. Although it may be interesting to find multiple subgroups, we note that it is not our main objective. We here focus on developing a method that can: 1) deal with arbitrary target distributions, 2) find symbolic and highly interpretable descriptions of numeric features in a differentiable end-to-end fashion and 3) significantly reduce the runtime compared to state-of-the-art baselines.
> Nonetheless, our method can be extended in a straightforward manner to find diverse sets of subgroups! The main idea is to iteratively search for subgroups and extend the optimization objective with a regularizer of the form $KL(P_{Y|S} || P_{Y|S_{i-1}})$, described in Appendix C. This allows us to find subgroups that differ from the overall distribution and simultaneously differ from previously found subgroups.
> 4. The effect of quantisation is shown in the first experiment of the synthetic experiments (Feature Thresholding), where we vary the number of cut points for subgroup discovery (SD-$\mu$). The results are displayed in Figure 1.b. For a small number of cutpoints (coarse quantization), the performance is significantly worse than for a finer quantization. To match the performance of SyFlow, SD-$\mu$ requires 40 cutpoints, while the runtime increases by three orders of magnitude.

---

### Author Response · Authors · 2023-11-22

We wish to express our gratitude to all reviewers for their time and providing useful feedback. You all agree on the relevance of finding subgroups with exceptional target distributions, especially for medical applications. In particular, you appreciated
1. our novel combination of neural rule induction and Normalising Flows, which permits learning subgroups in high-dimensional feature and target domains,
2. the ability of SyFlow to deal with arbitrary target distributions, as also showcased in the real-world datasets,
3. the scalability of SyFlow both in terms of the number of features and the number of samples.

As it seems to have been a common misunderstanding, we want to clarify the definition of the subgroup membership indicator S. S depends (solely) on the features X and the rule that defines it, and is (hence) not a latent variable of X. The confusion probably stems from the fact that we model the probability of a sample to be part of the subgroup. That is, during learning samples are permitted to be partial members of the subgroup, which is e.g. appropriate for example at the boundaries of the intervals. By decreasing the temperature, we obtain increasingly crisp membership of $P(S=1|x) \to 0$ or 1 so that S is asymptotically always 0 or 1 at a given x.

Multiple reviewers suggested that we investigate learning multiple subgroups. To accommodate this, while avoiding redundant results, we propose an additional regularization term that rewards (KL) divergence between the new subgroup and already found subgroups. We describe this regularizer in Appendix C of the updated manuscript.

Finally, we updated the experiment section in the manuscript and provided a description of the baseline methods.

---

### Meta-Review · Area_Chair_R7UV · 2023-12-12

**Metareview:**

This paper has been assessed by five knowledgeable reviewers, two of whom recommended straight rejection, two others weak rejection, and one weak acceptance. The authors engaged the reviewers in a discussion and provided a revised version of the manuscript, however those means have not changed the prevalently negative sentiment of the reviewers. The key remaining issues predominantly involve mathematical notation and clarity, and treatment of uncertainty in experiments that precludes proper assessment of statistical significance of the results. It is clear that this paper in its current form is not fit for inclusion in ICLR 2024.

**Justification For Why Not Higher Score:**

Clear rejection.

**Justification For Why Not Lower Score:**

N/A

---

### Decision · Program_Chairs · 2024-01-16

Reject